# In Vivo and In Vitro Enhanced Tumoricidal Effects of Metformin, Active Vitamin D_3_, and 5-Fluorouracil Triple Therapy against Colon Cancer by Modulating the PI3K/Akt/PTEN/mTOR Network

**DOI:** 10.3390/cancers14061538

**Published:** 2022-03-17

**Authors:** Riyad Adnan Almaimani, Akhmed Aslam, Jawwad Ahmad, Mahmoud Zaki El-Readi, Mohamed E. El-Boshy, Abdelghany H. Abdelghany, Shakir Idris, Mai Alhadrami, Mohammad Althubiti, Hussain A. Almasmoum, Mazen M. Ghaith, Mohamed E. Elzubeir, Safaa Yehia Eid, Bassem Refaat

**Affiliations:** 1Department of Biochemistry, Faculty of Medicine, Umm Al-Qura University, Al Abdeyah, Makkah 24381, Saudi Arabia; ramaimani@uqu.edu.sa (R.A.A.); mzreadi@uqu.edu.sa (M.Z.E.-R.); mathubiti@uqu.edu.sa (M.A.); mezubier@uqu.edu.sa (M.E.E.); syeid@uqu.edu.sa (S.Y.E.); 2Laboratory Medicine Department, Faculty of Applied Medical Sciences, Umm Al-Qura University, Al Abdeyah, P.O. Box 7607, Makkah 24381, Saudi Arabia; maaslam@uqu.edu.sa (A.A.); jaahmad@uqu.edu.sa (J.A.); elboshi@mans.edu.eg (M.E.E.-B.); ahahassan@uqu.edu.sa (A.H.A.); siidris@uqu.edu.sa (S.I.); haamasmoum@uqu.edu.sa (H.A.A.); mmghaith@uqu.edu.sa (M.M.G.); 3Biochemistry Department, Faculty of Pharmacy, Al-Azhar University, Assuit 71524, Egypt; 4Clinical Pathology Department, Faculty of Veterinary Medicine, Mansoura University, Mansoura 35516, Egypt; 5Department of Anatomy, Faculty of Medicine, Alexandria University, Alexandria 21544, Egypt; 6Department of Pathology, Faculty of Medicine, Umm Al-Qura University, Al Abdeyah, Makkah 24381, Saudi Arabia; mhhadrami@uqu.edu.sa

**Keywords:** chemoresistance, calcitriol, metformin, cell cycle, apoptosis, phosphatidylinositol-3-kinase, protein kinase B, mammalian target of rapamycin

## Abstract

**Simple Summary:**

Failure of chemotherapy is common during the treatment of colon cancer, and there is a compelling need to develop alternative therapeutic approaches against this common malignancy. Metformin, which is an oral hypoglycaemic agent used for treating diabetes mellitus, and vitamin D have shown promising anticancer activities, and both agents boosted the actions of chemotherapy against colon cancer. This study, therefore, measured the potential beneficial effects of adding metformin and/or active vitamin D to the main cytotoxic drug used for treating colon cancer. The results demonstrate that metformin had superior anticancer effects relative to active vitamin D and ameliorated the effects of chemotherapy in animals and in cells. To the best of our knowledge, this study is also the first to report that triple treatment with the drugs of interest showed the best inhibition of cancer progression, which could provide a better therapeutic strategy against colon cancer.

**Abstract:**

Chemoresistance to 5-fluorouracil (5-FU) is common during colorectal cancer (CRC) treatment. This study measured the chemotherapeutic effects of 5-FU, active vitamin D_3_ (VD_3_), and/or metformin single/dual/triple regimens as complementary/alternative therapies. Ninety male mice were divided into: negative and positive (PC) controls, and 5-FU, VD_3_, Met, 5-FU/VD_3_, 5-FU/Met, VD_3_/Met, and 5-FU/VD_3_/Met groups. Treatments lasted four weeks following CRC induction by azoxymethane. Similar regimens were also applied in the SW480 and SW620 CRC cell lines. The PC mice had abundant tumours, markedly elevated proliferation markers (survivin/CCND1) and PI3K/Akt/mTOR, and reduced p21/PTEN/cytochrome C/caspase-3 and apoptosis. All therapies reduced tumour numbers, with 5-FU/VD_3_/Met being the most efficacious regimen. All protocols decreased cell proliferation markers, inhibited PI3K/Akt/mTOR molecules, and increased proapoptotic molecules with an apoptosis index, and 5-FU/VD_3_/Met revealed the strongest effects. In vitro, all therapies equally induced G1 phase arrest in SW480 cells, whereas metformin-alone showed maximal SW620 cell numbers in the G0/G1 phase. 5-FU/Met co-therapy also showed the highest apoptotic SW480 cell numbers (13%), whilst 5-FU/VD_3_/Met disclosed the lowest viable SW620 cell percentages (81%). Moreover, 5-FU/VD_3_/Met revealed maximal inhibitions of cell cycle inducers (CCND1/CCND3), cell survival (BCL2), and the PI3K/Akt/mTOR molecules alongside the highest expression of cell cycle inhibitors (p21/p27), proapoptotic markers (BAX/cytochrome C/caspase-3), and PTEN in both cell lines. In conclusion, metformin monotherapy was superior to VD_3_, whereas the 5-FU/Met protocol showed better anticancer effects relative to the other dual therapies. However, the 5-FU/VD_3_/Met approach displayed the best in vivo and in vitro tumoricidal effects related to cell cycle arrest and apoptosis, justifiably by enhanced modulations of the PI3K/PTEN/Akt/mTOR pathway.

## 1. Introduction

Colorectal cancer (CRC) is the third most common malignancy and the fourth leading cause of cancer-related deaths worldwide [1,2]. Colon oncogenesis involves abnormal upregulations in the cyclin D1 (CCND1), CCND3, B-cell lymphoma 2 (BCL2), and survivin proteins that promote cell cycle progression and cell survival [3,4]. In contrast, colon carcinogenesis is linked with reductions in cyclin-dependent kinase (CDK) inhibitors (p21 and p27), BCL2-associated X protein (BAX), cytochrome C (Cyto-C), and caspase-3 (Casp-3) proteins [5,6,7]. The main drug used for treating advanced CRC, 5-fluorouracil (5-FU), inhibits cancer progression by increasing the expression of CDK inhibitors and several proapoptotic molecules [8,9]. However, 5-FU has limited therapeutic efficiency and is associated with low success and survival rates due to the development of resistance by neoplastic enterocytes [10,11].

Phosphatidylinositol-3-kinase (PI3K) is a cell membrane-associated kinase, and its class IA, which is a heterodimer of the p85 regulatory and p110 catalytic subunits, promotes the development and progression of many cancers, including CRC [12,13]. Mammalian target of rapamycin (mTOR) is a serine/threonine kinase that acts as a downstream effector for PI3K and is involved in the regulation of cell metabolism, growth, and survival [14,15]. The activation of PI3K is induced by phosphorylation of the p85α subunit that activates mTOR through the intracellular mediator, protein kinase B (Akt), and the hyperactivated PI3K/Akt/mTOR pathway is linked with CRC progression and resistance to chemotherapy [12,13]. In contrast, the endogenous inhibitor of the PI3K/Akt/mTOR pathway, phosphatase and tensin homolog (PTEN), is commonly lost during colon carcinogenesis [16,17]. Moreover, the use of specific inhibitors for PI3K/Akt/mTOR has shown promising anticancer effects [16,17].

Interestingly, many non-chemotherapeutic drugs exhibited potent anticancer activities against CRC by modulating the cell cycle regulatory and proapoptotic molecules alongside inhibiting the PI3K/Akt/mTOR pathway. In this regard, several studies revealed negative correlations between the levels of serum vitamin D and CRC development [18,19]. Moreover, active vitamin D_3_ (VD_3_) attenuated the expression of the PI3K/Akt/mTOR pathway, inhibited cell proliferation, and promoted apoptosis in colon cancer cell lines [20,21,22,23]. Other studies have also reported potent antitumorigenic effects for metformin against CRC that were portrayed by cell cycle arrest, increased apoptosis, and downregulations in the PI3K/Akt/mTOR network [24,25,26,27,28]. Additionally, others have shown enhanced tumoricidal actions against CRC by combining VD_3_ or metformin with 5-FU [21,29,30,31], as well as by adding both agents together [32,33].

However, none of the earlier studies explored the potential anticancer effects of 5-FU, VD_3_, and metformin triple therapy against CRC. Hence, this study measured the tumoricidal effects related to cell cycle progression, apoptosis, and inhibition of the PI3K/Akt/mTOR pathway following 5-FU, VD_3_, and/or metformin single, dual, and triple therapies in vivo and in vitro.

## 2. Materials and Methods

### 2.1. Chemicals and Reagents

Azoxymethane (AOM; #A5486-100MG) of 98% purity was obtained from Sigma-Aldrich Co. (St. Louis, MO, USA), whilst both active VD_3_ (Calcitriol; #HY-10002) and metformin (#HY-17471A) were from MedChemExpress LLC (Princeton, NJ, USA). Moreover, 5-FU was purchased from Hospira Australia Ltd. (Melbourne, Australia), and all drugs were freshly prepared as per the manufacturers’ instructions prior to their use. DMEM media (#10566032), foetal bovine serum (FBS; #A3160802), antibiotic-antimycotic solution (#15240062), and all the utilised cell culture materials were from Thermo Fisher scientific (Fremont, CA, USA).

### 2.2. Experimental Animal Studies and Treatment Protocols

Ninety male BALB/c mice, weighing 25–30 g each and of 12 weeks of age, were equally divided into nine groups (10 mice/group) as follows: the negative (NC) and positive (PC) controls; the 5-FU (5-FU), calcitriol (VD_3_), and metformin (Met) monotherapy groups; the dual therapy groups that simultaneously received VD_3_/5-FU (VF), Met/5-FU (MF), or VD_3_/Met (VM); and the triple therapy group (VMF) that was treated with all drugs. All animal experiments were conducted in compliance with the European guidelines for the care and use of laboratory animals, and the study was approved by the Committee for the Care and Use of Laboratory Animals at Umm Al-Qura University (AMSEC 19/05-10-20).

Following one week of acclimatisation, AOM was injected intraperitoneally for two consecutive weeks (10 mg/kg/week) in all groups, except the negative control mice, to induce CRC without exceeding the tumour size/burden to ensure the welfare of animals, as previously reported [21]. The animals were then observed for another 20 weeks with no intervention and received standard laboratory chow with water ad libitum. All treatment protocols were initiated at week 21 post-AOM and continued for 4 weeks. Freshly prepared calcitriol (0.07 µg/kg/day; five times/week) and metformin (430 mg/kg/day; five times/week) were administrated orally to the designated groups, and the delivered amounts were equivalent to the highest daily recommended doses for a 60 kg body weight adult human for calcitriol (0.25 µg/day; 0.0042 µg/kg/day) [34] and metformin (1500 mg/day; 25 mg/kg/day) [35] as per the dose conversion formula between humans and mice [36]. 5-FU was administrated to the designated groups for four successive cycles as a single weekly intraperitoneal injection (50 mg/kg/week) as previously described [21]. The study design is summarised in Appendix A. Euthanasia was performed on the first day of week 25 post-AOM by cervical dislocation under anaesthesia, as described earlier [21]. Blood samples were collected from each animal, and serum was stored at −20 °C until use to measure liver and renal biochemical parameters. The colon from each animal was removed, flushed with cold phosphate-buffered saline (PBS), cut longitudinally, and soaked in 10% formalin overnight between layers of filter paper [37].

#### 2.2.1. Gross Examination and Tumour Counting

The numbers of tumours/colon were grossly counted the next morning by two researchers, followed by cutting each colon into three equal parts representing proximal, middle, and distal segments to the caecum [37]. All parts were stained for 10 min by 1% Alcian blue solution (#sc-214517; Santa-Cruz Biotechnology Inc.; Dallas, TX, USA) in 3% acetic acid (pH 2.5) and then examined under a dissecting microscope (Human Diagnostics; Wiesbaden, Germany) to identify mucin-depleted foci (MDF) and to count the small tumours that were not detected by the naked eye [37].

Each colonic segment was halved longitudinally, and a specimen was Swiss rolled and processed by conventional histopathology techniques. Total RNA was extracted from the residual colonic samples with tumours by PureLink^TM^ RNA Mini Kit (#12183025; Thermo Fisher Scientific; Fremont, CA, USA), whilst total proteins were extracted by homogenising colonic specimens in RIPA lysis buffer (#89900) with protease inhibitors (#78429; Thermo Fisher Scientific). While the RNA quality and quantities were measured with a Qubit4 Fluorometer (Thermo Fisher Scientific), the concentrations of extracted total protein were measured by a Pierce™ Rapid Gold BCA Protein Kit (#A53225; Thermo Fisher Scientific). The total protein samples were then diluted with deionised water (2000 µg/mL) to be used for the enzyme-linked immunosorbent assay (ELISA).

#### 2.2.2. Histopathological Studies of Colon Tissues

Tissue sections of 5 μm thickness from each colon were stained by H&E and examined on a Leica DMi8 microscope (Leica Microsystems, Wetzlar, Germany) at 100× and 200× magnifications by two expert researchers who were blind to the source group. Both examiners reported the histological features of adenocarcinomas in five random fields from each section using a set of well-established criteria [37]. In case of a wide disagreement between both examiners, an independent expert histopathologist re-examined the sections. The captured images were analysed by the ImageJ software (https://imagej.nih.gov/ij/ (accessed on 6 December 2021)) to measure the surface areas (μm^2^) of adenocarcinomas, as reported earlier [21,37].

#### 2.2.3. Immunohistochemistry (IHC)

All the primary antibodies were from Cell Signaling Technology Inc. (Danvers, MA, USA). Rabbit monoclonal antibodies were used to detect CCND1 (#55506), CDK inhibitor-1A (p21; #37543), and mTOR (#2983), whereas PI3K-p85α (#13666) was detected by mouse monoclonal IgG antibodies. An avidin-biotin horseradish peroxidase technique was applied on 5 μm tissue sections to detect the targeted proteins as previously described [38]. Briefly, BLOXALL^®^ Solution (#SP-6000-100; Vector Laboratories Inc.; Burlingame, CA, USA) was used to block endogenous peroxidases. The primary antibodies (1:200 concentration for all) were added, and the slides were incubated at 4 °C overnight. The following morning, ImmPRESS^®^ HRP Horse Anti-mouse (#MP-7402) or anti-rabbit (#MP-7401) IgG Plus Polymer Peroxidase Kits were used as per the manufacturer’s guidelines (Vector Laboratories Inc.). An identical protocol was applied for the negative control slides, but primary isotype mouse (#sc-2025) or rabbit (#sc-2027) IgG antibodies (Santa-Cruz Biotechnology Inc.) were used to control for non-specific staining. The sections were counterstained with haematoxylin and examined with a 20× objective on a Leica DMi8 microscope. Digital images were acquired from 10 different fields/section, and the protein expression was analysed by ImageJ software using the IHC Image Analysis Toolbox as previously reported [38,39].

#### 2.2.4. Terminal Deoxynucleotidyl Transferase-dUTP Nick End Labelling (TUNEL) Assay

Cell apoptosis was detected in colonic tissues with a Click-iT™ Plus TUNEL Assay (#C10617; Thermo Fisher Scientific) using the manufacturer’s protocol. The apoptotic bodies were co-localised with cleaved Casp-3 by applying a sequential protocol as reported earlier [40,41]. After completing the TUNEL protocol, anti-cleaved Casp-3 rabbit IgG monoclonal antibodies (#9661; Cell Signaling Technology Inc.; Danvers, MA, USA) were added at a 1:400 concentration, and the slides were incubated for 3 h. Donkey anti-rabbit IgG antibodies tagged with Alexa Fluor™ 555 (#A-31572; Thermo Fisher Scientific) were then added for 30 min followed by DAPI counterstaining (#D3571; Thermo Fisher Scientific). The slides were cover-slipped with a permanent fluorescence mounting medium (#S3023; Dako, CA, USA) and observed on a Leica DMi8 microscope at 400× magnification. Images were acquired from 15 non-overlapping fields/section, and the apoptotic cells’ numbers and cleaved Casp-3 staining intensity were measured by ImageJ software as previously reported [39,42].

#### 2.2.5. Liver and Renal Biochemical Profiles

Liver enzymes (ALT, ALP, and AST) and kidney function parameters (creatinine/urea, and total protein) were measured in serum samples using a Cobas e411 machine and by following the manufacturer’s protocols (Roche Diagnostics, Mannheim, Germany).

#### 2.2.6. ELISA

The colonic tissue concentrations of survivin (#SEC045Mu), cytochrome C (#SEA594Mu), Akt1 (#SEC231Mu), and PTEN (#SEF822Mu) were measured by specific mouse ELISA kits (Cloud-Clone Corp.; Houston, TX, USA). The samples were processed in duplicate on an automated ELISA machine (Human Diagnostics, Wiesbaden, Germany) according to the manufacturer’s protocols.

### 2.3. Cell Culture and Treatment Protocols

The human SW480 primary and SW620 metastatic colon cancer cell lines were purchased from the American Type Culture Collection (ATCC; Manassas, VA, USA). The SW480 and SW620 cells were cultured in DMEM that contained 10% FBS and 1% antibiotic-antimycotic solution. All cells were grown in a humified incubator at 37 °C and 5% CO_2_.

The 5-FU (50 µM), VD_3_ (25 µM), and metformin (39.8 mM) concentrations (IC50) were determined by the 3-(4,5-Dimethylthiazol-2-yl)-2,5-Diphenyltetrazolium Bromide (MTT) cytotoxicity assay at 24 h (Appendix A). For cell cycle analysis, the SW480 (2 × 10^5^) and SW620 (3 × 10^5^) cells were seeded in 6-well plates for 24 h and then treated for 12 h with 5-FU, VD_3_, and/or metformin single, dual, and triple therapies, resulting in the following groups: untreated control (Ctr); 5-FU, VD_3_, and Met monotherapies; VF, MF, and VM co-therapies; and VMF triple therapy. The 12 h timepoint was applied to assure that any effects of combination therapies could be accurately analysed by cell cycle, apoptosis, gene, and protein expression techniques.

#### 2.3.1. Cell Cycle Analysis

Cell cycle analysis was performed following the different treatment regimens in the SW480 and SW620 cells as previously described [21]. Briefly, cells were washed twice with PBS (500× *g* for 5 min) following trypsinisation and fixed in ice-cold 70% ethanol for 24 h at 4 °C. The cells were washed twice in PBS (600× *g* for 5 min) and treated for 15 min with RNase A (20 µg/mL; #12091021; Thermo Fisher, Fremont, CA, USA), and 2 µg/mL propidium iodide (PI; #P1304MP; Thermo Fisher) was added. Following staining, cell cycle analysis was immediately performed with a Novocyte 3000 flow cytometer (Agilent Technologies, Santa Clara, CA, USA). The percentages of cells in different phases of the cell cycle (Sub-G1, G0/G1, S, G2/M) were determined for 20,000 acquired events using the NovoExpress software cell cycle algorithm, and data are shown as the mean ± SD (n = 3).

#### 2.3.2. Apoptosis Assay

The Annexin V-FITC/PI Apoptosis Assay Kit (#V13245; Thermo Fisher Scientific) was used according to the manufacturer’s protocol. The cells were harvested after the different treatment protocols, washed twice with ice-cold PBS, and re-suspended in 100 µL of 1× Annexin V (AV) binding buffer. For cell staining, the mixture of AV-FITC (5 µL) and PI (1 µL) was added to each 100 µL of the SW480 and SW620 cell suspensions followed by incubation in the dark for 15 min at room temperature. Subsequently, the AV binding buffer (400 µL) was added, and the cells were placed on ice and immediately analysed using the NovoCyte 3000 flow cytometer. The experiments were processed in triplicate, and the data represent the percentage (mean ± SD) of the different apoptosis stages as follows: live (unstained), early (AV+/PI−) and late apoptotic (AV+/PI+), and dead (AV−/PI+) cells.

#### 2.3.3. Quantitative Reverse-Transcription Polymerase Chain Reaction

Following 12 h treatment with the different therapies, the SW480 and SW620 cells were trypsinised and washed in PBS. Total RNA was extracted by a PureLink^TM^ RNA Mini Kit (Thermo Fisher Scientific), and a high-capacity Reverse Transcription Kit was used for cDNA synthesis (#4368814; Thermo Fisher Scientific). PCR was performed in triplicate wells/sample and consisted of 40 amplification cycles (95 °C/15 s and 60 °C/1 min) that were processed with a QuantStudio™ 3 System. Each well contained 5 µL SYBR Green, 3 µL nuclease-free water, 5 pmol (1 µL) of each set of primers (Appendix A), and 25 ng cDNA (1 µL). A minus-reverse transcription control from the prior RT step and a separate minus-template PCR, in which the cDNA was replaced by nuclease-free water, were used as negative controls. The relative expression of *CCND1*, *CCND3*, *p21*, *p27*, *BCL2*, *BAX*, *Cyto-C*, *caspase-3*, *PIK3CA*, *PTEN*, *AKT1*, and *mTOR* genes was calculated by the 2^−∆∆Ct^ method [43].

#### 2.3.4. Western Blot

All primary antibodies were from Cell Signaling Technology Inc. (MA, USA). While mouse monoclonal IgG antibodies were used to detect CCND3 (#2936), BCL2 (#15071), and PI3K-p85α, rabbit monoclonal antibodies were used to detect CCND1, p21, p27 (#3686), BAX (#2772), Cyto-C (#4272), cleaved Casp-3, PTEN (#9188), Akt1 (#75692), and mTOR proteins by Western blotting. GAPDH loading control mouse monoclonal antibodies (#MA5-15738-1MG; Thermo Fisher Scientific) were used for normalisation.

Briefly, total proteins were extracted from each cell pellet, and 50 µg total protein from each sample was loaded on gradient 4–20% Mini-PROTEAN^®^ TGX Stain-Free™ SDS-PAGE gels (#4568096; Bio-Rad Laboratories Inc.; Hercules, CA, USA). The resolved proteins were then transferred to 0.2 µm Trans-Blot^®^ Turbo^TM^ PVDF membranes using a Trans-Blot^®^ Turbo^TM^ Transfer System (Bio-Rad Laboratories Inc.; Hercules, CA, USA). The membranes were blocked for 15 min with SuperBlock^TM^ T20 buffer (TBS-T; #37543; Thermo Fisher Scientific) and incubated with primary antibodies (1:1000 for all antibodies) overnight at 4 °C. Next, the membranes were washed with TBS-T and incubated for 1 h with WestVision™ secondary anti-mouse (#WB-2000-.8) or anti-rabbit (#WB-1000-.8) IgG antibodies (1:10,000) conjugated with peroxidase micropolymer (Vector Laboratories Inc.; Burlingame, CA, USA). The membranes were washed, and the signals were developed by SignalFire^™^ Plus ECL Reagent (#12630; Cell Signaling Technology Inc.; Danvers, MA, USA). The images were acquired by a ChemiDocTM XRS+ (Bio-Rad Laboratories Inc.), and the density of each protein band was quantified and then normalised against the densitometry of the GAPDH band by ImageJ software (https://imagej.nih.gov/ij/ (accessed on 6 December 2021)) as previously described [44]. Data are shown as the mean ± SD of three blots/cell line for each targeted protein. Original images for Western blotting are shown in Appendix A.

### 2.4. Statistical Analysis

SPSS statistical analysis software version 25 was used to analyse the data, and all variables were assessed for normality and homogeneity by the Kolmogorov–Smirnov test and the Levene test, respectively. One-way analysis of variance (ANOVA) followed by Tukey’s HSD or Games–Howell post hoc tests was performed to compare between the study groups based on variance equality. Significance was considered when the *p* value was <0.05.

## 3. Results

### 3.1. Effects of Treatment Regimens on CRC Progression in Mice

None of the animals died during the study, and the results of liver and renal function parameters were comparable between the different study groups (Appendix A). Colonic tissues from the NC group demonstrated a normal architecture under the dissecting microscope as well as normal histology following H&E staining (Figure 1A). In contrast, numerous MDF were seen in the PC group colons and co-existed with large numbers of gross and microscopic tumours. The bright-field microscope also revealed abundant colonic adenocarcinomas in the PC group that had large surface areas with poor to moderate differentiated histology (Figure 1).

The numbers of MDF (Figure 1B), dissecting microscope tumours (Figure 1D), total numbers of tumours (Figure 1E), and the surface areas of adenocarcinomas (Figure 1G) decreased significantly and equally in all monotherapy groups relative to the PC group. Moreover, the numbers of gross tumours were markedly lower in the 5-FU and Met groups, but not in the VD_3_ group, compared with the PC mice. Nevertheless, the numbers of adenocarcinomas under a light microscope were comparable between the PC and all monotherapy groups (Figure 1F).

All dual therapy protocols showed markedly lower numbers of MDF, gross and microscopic tumours, and adenocarcinomas alongside smaller adenocarcinoma surface areas compared with the PC, 5-FU, VD_3_, and Met groups. However, the antitumorigenic effects were significantly more pronounced in the MF group relative to the VF and VM co-therapy groups (Figure 1). On the other hand, the lowest numbers of MDF, gross tumours, and adenocarcinomas were detected in the triple therapy (VMF) group colons compared with the PC group as well as all single and dual therapy groups (Figure 1).

### 3.2. Effects of the Different Treatment Protocols on Cell Cycle

#### 3.2.1. In Vivo Expression of CCND1 and p21

While the gene and protein expression of CCND1 increased significantly, the p21 mRNA and protein diminished in the PC colonic tissues compared with the NC group (Figure 2). All monotherapies equally demonstrated marked declines in the mRNA and protein of CCND1 with concurrent significant increases in p21 gene and protein expression relative to the PC colonic tissues. Additional significant inhibitions in the CCND1 gene and protein expression alongside marked upregulations in the p21 gene and protein were detected in all co-therapy groups relative to the monotherapy groups (Figure 2). Moreover, the CCND1 gene and protein expression was significantly lower and coincided with markedly higher p21 mRNA and protein in the MF group than the VF and VM co-therapies (Figure 2). Nonetheless, the highest p21 gene and protein expression together with the lowest levels of CCND1 mRNA and protein was observed in the VMF group compared with the PC, 5-FU, VD_3_, Met, VF, MF, and VM groups (Figure 2).

#### 3.2.2. In Vitro Cell Cycle Arrest and Expression of Cell Cycle Regulatory Molecules

Single treatment with 5-FU and metformin equally and markedly augmented the SW480 cell counts in the Sub-G1 phase (2-fold, for both), and the highest increases were seen in the VM group (2.4-fold) compared with non-treated cells (Figure 3a). Alternatively, all monotherapies showed equal increases in the SW620 cell numbers in the Sub-G1 phase (1.3-fold, for all), whilst the MF co-therapy revealed the maximal increase (2.8-fold) relative to control cells (Figure 3b). All treatments, except VD_3_ monotherapy, markedly increased the SW480 cell numbers in the G0/G1 phase (~1.5-fold) compared with untreated cells, and the results were comparable between the different therapies (Figure 3a). Although all treatments were also associated with G0/G1 phase arrest in the SW620 cells, metformin monotherapy showed the maximal significant increase relative to non-treated cells (1.6-fold; Figure 3b).

All monotherapies also revealed marked decreases in CCND1 and CCND3 alongside increases in the p21 and p27 genes (Figure 4a) and proteins (Figure 4b–f) relative to untreated SW480 and SW620 cells, and the effects of metformin-alone were more prominent in both cell lines. The dual therapies further downregulated CCND1 and CCND3 whilst promoting p21 and p27 genes and proteins compared with all monotherapies in both cell lines, and the MF approach was markedly more effective relative to the other dual therapies. However, the lowest CCND1 and CCND3 alongside the highest upregulations in the p21 and p27 genes (Figure 4a) and proteins (Figure 4b–f) were detected with the triple therapy in both cell lines compared with untreated cells as well as with the other therapeutic approaches.

### 3.3. Effects of the Different Treatment Protocols on Cell Death and Apoptosis Markers

#### 3.3.1. Colonic Cell Apoptosis and Apoptosis Markers In Vivo

In the NC group colonic tissues, apoptotic bodies and cleaved Casp-3 protein were mainly detected in glandular, luminal, and cryptic epithelia together with stromal cells (Figure 5A). The numbers of apoptotic bodies, cleaved Casp-3 protein, colonic tissue levels of the Cyto-C protein, and the apoptosis index decreased substantially in the PC mice and concurred with a marked increase in the survivin protein relative to the NC group (Figure 5). All monotherapies revealed marked declines in colonic tissue survivin concentrations alongside increases in colonic tissue Cyto-C levels with a higher apoptosis index and Casp-3 protein expression compared with the PC group. While the expression of the Casp-3 protein and the apoptosis index were significantly higher in the 5-FU group than the VD_3_ and Met groups (Figure 5B), colonic survivin and Cyto-C tissue concentrations were similar between all monotherapies (Figure 5C). Moreover, cleaved Casp-3 and the apoptosis index were markedly higher in the Met group than the VD_3_ group (Figure 5B).

Although colonic Cyto-C, cleaved Casp-3, and the apoptosis index were markedly higher in all co-therapy groups and coincided with significant reductions in survivin relative to the PC and monotherapy groups, the results were significantly stronger in the MF group than the VD and VM groups (Figure 5). On the other hand, the VMF group demonstrated the maximal cyto-C concentrations, the highest apoptosis index and cleaved Casp-3 expression, and the lowest survivin levels compared with the PC, monotherapy, and dual therapy groups (Figure 5).

#### 3.3.2. In Vitro Cell Apoptosis and Expression of Apoptosis Markers

All treatment protocols significantly reduced the numbers of living cells in the cell lines used relative to untreated cells, except for the VD_3_ single treatment that showed comparable results to the SW620 control cells (Figure 6). In the SW480 cells, 5-FU with metformin dual treatment displayed the highest rate of cell death that was depicted with marked increases in the numbers of cells in early and late apoptosis relative to all therapeutic protocols (Figure 6a). On the other hand, the minimal numbers of viable SW620 cells were detected in the VMF group that concord with the maximal increases in the percentages of early and late apoptotic cells compared with all groups (Figure 6b).

The monotherapies also revealed marked decreases in BCL2 along with increases in the BAX, Cyto-C, and Casp-3 genes (Figure 7a) and proteins (Figure 7b–f) compared with untreated SW480 and SW620 cells, and the metformin single therapy was more prominent relative to all monotherapy groups. In contrast, and although the dual therapies also reduced the expression of BCL2 while increasing that of the BAX, Cyto-C, and Casp-3 genes and proteins in both cell lines, the MF co-therapy was the most efficient compared with the other dual therapies. Moreover, the triple therapy regimen showed the best downregulation in the mRNAs (Figure 7a) and proteins (Figure 7b–f) of BCL2 along with the maximal increases in BAX, Cyto-C, and Casp-3 relative to all therapies in the tested cell lines.

### 3.4. Effects of the Different Treatment Protocols on the Expression of PI3K/Akt/mTOR Oncogenic Pathway

#### 3.4.1. Protein Expression of PI3K/Akt/mTOR in Colonic Tissues

The PI3K-p85α and mTOR protein expression alongside the tissue levels of the Akt1 protein increased significantly, whereas PTEN protein concentrations declined, in the PC colonic tissues relative to the NC group (Figure 8). Although 5-FU monotherapy significantly reduced the expression of the PI3K-p85α protein compared with the PC group, the levels of the Akt1, PTEN, and mTOR proteins were comparable between both groups. Single treatment with metformin and VD_3_ also showed substantial reductions in PI3K-p85α and Akt1 alongside increases in the PTEN protein compared with the PC group. However, metformin monotherapy showed significantly more effective modulatory effects on the targeted proteins relative to the 5-FU and VD_3_ groups (Figure 8). All dual therapy protocols showed further significant declines in PI3K-p85α, Akt1, and mTOR with concurrent increases in the PTEN protein compared with all monotherapy groups, and the MF group was significantly more effective than the other co-therapy groups. However, the triple therapy regimen showed the lowest PI3K-p85α, Akt1, and mTOR proteins alongside the highest concentrations of the PTEN protein compared with the PC group and all single and dual treatment protocols (Figure 8).

#### 3.4.2. In Vitro Gene and Protein Expression of the PI3K/Akt/mTOR Oncogenic Pathway

The monotherapy groups disclosed significant inhibitions in the PI3K-p85α, Akt1, and mTOR genes (Figure 9a) and proteins (Figure 9b–f) alongside increases in the PTEN mRNA and protein in the SW480 cells, and metformin monotherapy showed the most significant modulations compared with untreated cells and the other single therapies. In the SW620 metastatic colon cancer cells, however, 5-FU and VD_3_ monotherapies showed negligible effects on the targeted genes and proteins, whereas metformin exhibited marked modulations at the transcriptional and translational levels. Although the MF co-therapy approach showed the most prominent decreases in the genes (Figure 9a) and proteins (Figure 9b–f) of PI3K-p85α, Akt1, and mTOR with concurrent increases in the PTEN mRNA and protein relative to all dual therapy protocols, the maximal significant modulatory effects in both cell lines were observed with the triple therapy.

## 4. Discussion

This study compared the anticancer effects of single, dual, and triple treatments with 5-FU, active VD_3_, and/or metformin against CRC in vivo and in vitro. The gold standard chemotherapy used for treating CRC, 5-FU, provokes cell cycle arrest at the S phase by inhibiting thymidylate synthase, an enzyme required for DNA replication [8,9]. Others have also shown arrest at the G0/G1 phase of the cell cycle following 5-FU single therapy in the HCT116 and SW620 CRC cell lines [45,46]. Concurrently, 5-FU halts the progression of colon cancer by modulating many proapoptotic pathways [21,47]. However, cancerous cells often resist 5-FU actions through several mechanisms, including increasing the energy supply by switching to aerobic glycolysis to sustain survival, a phenomenon known as the Warburg effect [48]. Under this theme, the PI3K/Akt/mTOR signalling pathway is a central regulator of cell metabolism and survival, and during colon carcinogenesis, the pathway is abnormally hyperactivated, whilst the expression of its endogenous inhibitor, PTEN, declines [16,47,49]. The aberrant activation of the PI3K/Akt/mTOR network suppresses apoptosis by upregulating the cell survival molecule BCL2 that inhibits BAX and subsequently the release of Cyto-C from mitochondria [16,49,50]. The dysregulation in the PI3K/PTEN/Akt/mTOR pathway also promotes the metabolic shifting to glycolysis, thus reducing the efficacy of 5-FU by decreasing its absorption, due to acidification of the tumour microenvironment, as well as hindering cell apoptosis [16,49,50].

Herein, 5-FU monotherapy moderately decreased the numbers of MDF and tumours, but not adenocarcinomas, compared with the PC mice. Single treatment with 5-FU also induced cell cycle arrest at the G0/G1 phase in the SW480 and SW620 cells and was associated with marked increases in the numbers of apoptotic cells, both in vivo and in vitro. Moreover, 5-FU monotherapy significantly reduced the cell cycle-inducing (CCND1/CCND3) and survival (survivin/BCL2) markers alongside promoting the cell cycle inhibitory (p21/p27) and proapoptotic (BAX/Cyto-C/Casp-3) molecules at the gene and protein levels in treated mice as well as in SW480 cells. However, the cytotoxic drug showed limited efficacy on the expression of cell cycle and apoptosis regulatory molecules in the SW620 metastatic cells. Additionally, 5-FU single therapy showed negligible effects on the gene and protein expression of PI3K, Akt, PTEN, and mTOR in vivo and in vitro. Our results agree with many studies that have shown cell cycle arrest and apoptosis in colon cancer cells with 5-FU single treatment [8,9,21,45,46,47], and further support the notion that resistance to chemotherapy could, at least in part, be related to limited efficacy in modulating the PI3K/PTEN/Akt/mTOR pathway by 5-FU monotherapy [16,48,49].

Chemoresistance is a major clinical problem during the treatment of CRC, especially during the late stages of malignancy, and the quest for developing alternative and/or complementary therapies has therefore been the target of many studies [10,11]. In this context, the overall and cancer-specific survival rates were significantly higher in diabetic CRC patients who were using metformin relative to non-users [51,52]. Other in vitro studies have likewise reported persuasive antitumorigenic effects of metformin single therapy that were depicted by marked suppression of cell cycle progression, induction of apoptosis, and inhibition of the PI3K/Akt/mTOR molecules [24,25,26,27,28]. Similarly, several epidemiological studies disclosed inverse links between serum VD levels and the prevalence of CRC [18,19]. Active VD_3_ treatment also reduced cell proliferation, altered the mitochondrial membrane potential, inhibited the PI3K/Akt/mTOR pathway, and suppressed glycolysis alongside inducing cell cycle arrest and apoptosis in several human colon cancer cell lines [20,21,22,23]. Moreover, the chemopreventive actions of VD_3_ and metformin against CRC were compared by only two studies, and their results exhibited equal efficacies for both drugs in vitro [32] and in vivo [33], whereas their combination showed boosted antitumorigenic actions. Additionally, others have revealed enhanced anticancer effects against CRC by adding VD_3_ [21,29] or metformin [30,31] to 5-FU. However, none of the earlier studies compared the therapeutic efficacies between dual and triple therapies against CRC progression using metformin with VD_3_ and/or 5-FU.

This study revealed cell cycle arrest with increased p21 and p27 alongside decreased CCND1 and CCND3 genes and proteins following in vivo and in vitro treatment with VD_3_ and metformin monotherapies, confirming many reports that proclaimed suppression of the cell cycle by both agents [20,21,24,25,26,27,28]. However, in vivo and in vitro metformin single treatment also showed higher cell death, increased gene and protein expression of apoptosis markers and PTEN, and downregulations of the PI3K/Akt/mTOR molecules, whereas the VD_3_ monotherapy effects were weaker. While all dual therapy protocols disclosed enhanced modulations of the PI3K/PTEN/Akt/mTOR pathway, higher numbers of apoptotic cells, and a boosted expression of proapoptotic molecules compared with all monotherapies, the effects were markedly more pronounced with Met/5-FU co-therapy relative to the VD_3_/5-FU and Met/VD_3_ groups, both in vivo and in vitro.

The current data suggest that metformin monotherapy is superior to VD_3_ single therapy, and that metformin appears to be a better chemosensitiser to 5-FU cytotoxicity than VD_3_ for the treatment of CRC. A plausible explication for the disagreement between the present and the earlier in vivo and in vitro studies on the VD_3_ anticancer efficacy could be related to the relatively shorter treatment durations (4 weeks, and 12 h) in our study compared with the longer timepoints (≥8 weeks, and ≥24 h) in the prior reports [21,32,53,54]. Furthermore, it has also been suggested that daily intake of low-dose VD_3_ (400 IU) could trigger anticancer activities, whereas high doses (≥2000 IU) might promote cancer progression [55]. In contrast, others have shown that high dietary intake of VD_3_ significantly reduced the risk of CRC in individuals not receiving prescribed VD_3_ supplements [56]. Therefore, additional in vitro and in vivo studies are still needed to measure the anticancer activities of VD_3_ using low and high supplement doses [55,56].

We also hypothesise that the superiority of metformin’s proapoptotic effects over VD_3_, with and without 5-FU, could involve better attenuations of the PI3K/Akt/mTOR oncogenic molecules by inducing the expression of the tumour suppressor protein, PTEN, thus promoting oxidative metabolism and subsequently mitochondrial-induced apoptosis in neoplastic cells [24,25,26,27,28]. However, our data also reveal that the anticancer effects of the dual therapy protocols were cell-specific, since VM and MF co-therapies showed the highest increase in the SW480 and SW620 cells in the Sub-G1 phase, respectively. An explanation for this observation could be related to diversities in the molecular phenotypes of each of the cell lines used, which could have induced different anticancer responses due to molecular variations between the SW480 and SW620 cells [57,58,59].

Additionally, our data reveal that the VMF group had the highest expression of PTEN along with the cell cycle inhibitory and proapoptotic molecules that coincided with the lowest levels of CCND1, CCND3, BCL2, PI3K, Akt, and mTOR compared with all therapeutic regimens in the animals and the colon cancer cell lines used. To the best of our knowledge, this study is the first to demonstrate enhanced antitumorigenic effects of VD_3_/Met/5-FU triple therapy in the treatment of colon cancer, which could represent the best strategy for treating early and late stages of colon neoplasia. However, more studies that apply several timepoints (e.g., 12, 24, 48, and 72 h) and varying doses (low, intermediate, and high) of VD_3_ treatment, and that measure the markers of glycolysis and oxidative phosphorylation following the different therapeutic protocols, are needed to confirm our suggestions.

There are several drawbacks to the present study. Firstly, the AOM-induced CRC model is known to only mimic the early stages of colon malignancy, and other animal models (e.g., APC mutant mice) should be used in future studies to measure the effects of the drugs of interest against the advanced stages of CRC [60]. Moreover, we only measured the expression of PTEN, and other negative regulators of the PI3K/Akt/mTOR pathway, such as AMPK and GSK-3β, should be included in future studies, since both molecules have been shown to control cell death in malignant cells [61]. Additionally, we only applied concomitant combination protocols, and thus additional studies that use consecutive combinatory regimens (e.g., Met followed by VD_3_ and/or 5-FU) are needed to precisely identify the most effective approach for CRC treatment. The expression of enzymes involved in oxidative metabolism with the markers of glycolysis should also be measured in future studies to corroborate the present findings. Moreover, phosphorylated PI3K/Akt/mTOR proteins should be measured following different protocols, with and without applying specific inhibitors of this oncogenic molecular network, to fully elucidate the underlying mechanisms of the enhanced anticancer effects of the drugs of interest. Finally, the effects of the applied treatment protocols on the expression of Akt subtypes, including Akt1 and Akt2, should be investigated to explore their apoptotic and metabolic actions, since Akt1 is mainly involved in cell survival, whilst the Akt2 molecule regulates glucose homeostasis [62,63].

## 5. Conclusions

Metformin single therapy was superior to active VD_3_ in its anticancer effects, showing a higher expression of p21, p27, PTEN, BAX, Cyto-C, and Casp-3, inhibitions of CCND1, CCND3, BCL2, and the PI3K/Akt/mTOR network, and higher rates of apoptosis, both in vivo and in vitro. Although all dual therapy protocols revealed enhanced modulations of the PI3K/PTEN/Akt/mTOR pathway, alongside a higher expression of the cell cycle inhibitors and proapoptotic molecules than the monotherapies, the Met/5-FU co-therapy was better relative to the other dual therapies. In contrast, the triple therapy regimen exhibited the best anticancer effects related to cell cycle arrest and apoptosis compared with the single and dual protocols, possibly by boosted attenuations of the PI3K/Akt/mTOR oncogenic pathway. Nevertheless, more studies are needed to measure the tumoricidal effects of VD_3_ and/or metformin concomitant and sequential therapies, with and without 5-FU, using multiple timepoints, and to measure the metabolic markers of glycolysis and oxidative phosphorylation, in order to accurately determine their therapeutic values against CRC.

## Figures and Tables

**Figure 1 cancers-14-01538-f001:**
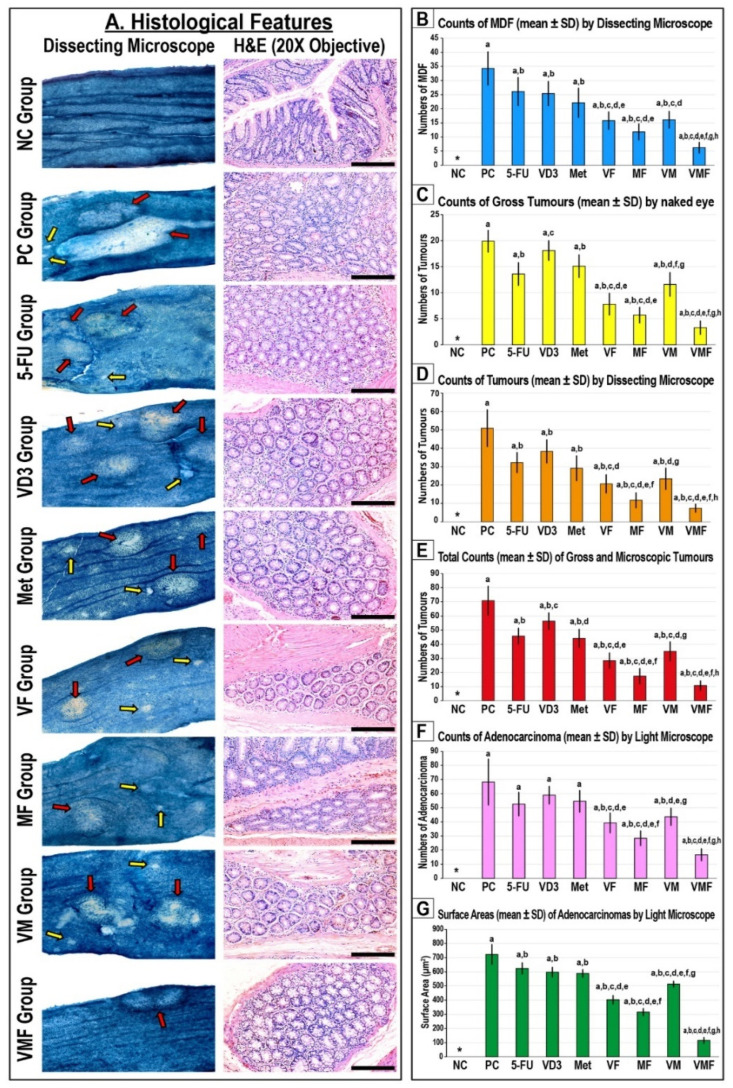
(**A**) Mouse colon mucosa from all the study groups under a dissecting microscope (n = 10 mice/group; ×20 magnification; red arrow = tumours; yellow arrow = mucin-depleted foci (MDF)) alongside colonic tissue sections from all groups by H&E stain (n = 10 mice/group; ×200 magnification; scale bar = 15 μm). Furthermore, the numbers of (**B**) MDF, (**C**) gross tumours, (**D**) microscopic tumours, (**E**) total tumours, and (**F**) adenocarcinomas alongside (**G**) the adenocarcinomas’ surface areas are shown as bar graphs (n = 10 mice/group; data were analysed by ANOVA followed by Games–Howell post hoc tests and are shown as the mean ± SD; * = not detected; a = *p* < 0.05 compared with the NC group; b = *p* < 0.05 compared with the PC group; c = *p* < 0.05 compared with 5-FU monotherapy; d = *p* < 0.05 compared VD_3_ monotherapy; e = *p* < 0.05 compared with Met monotherapy; f = *p* < 0.05 compared with VF dual therapy; g = *p* < 0.05 compared with MF dual therapy; and h = *p* < 0.05 compared with VM dual therapy). * = Not detected.

**Figure 2 cancers-14-01538-f002:**
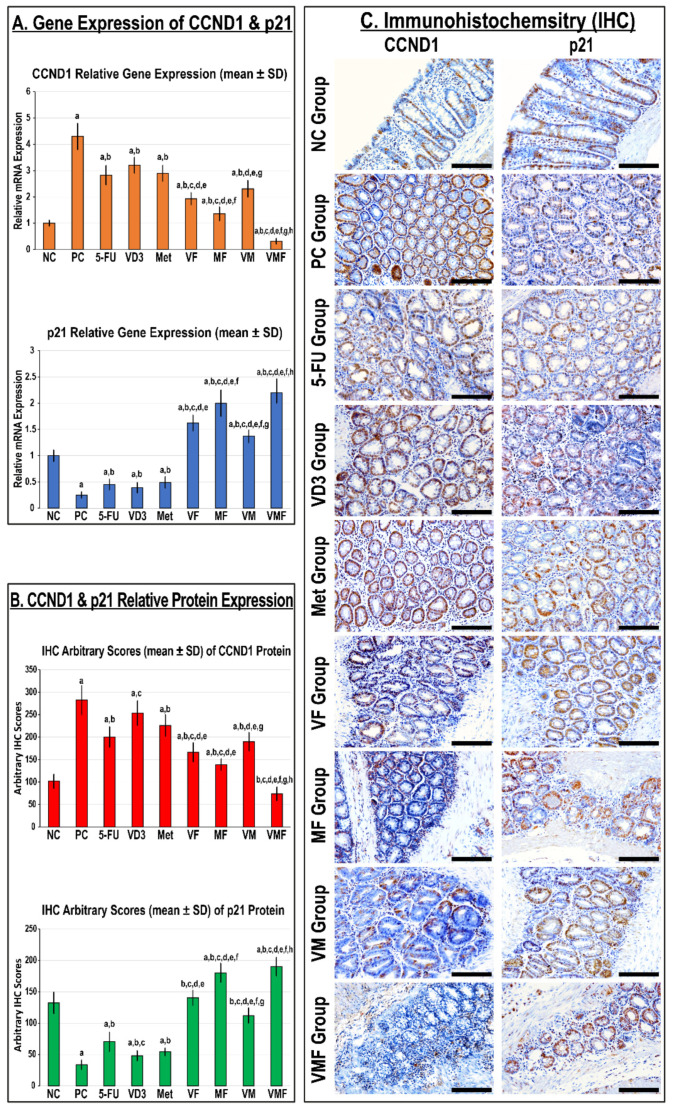
The relative (**A**) gene and (**B**) protein expression of CCND1 and p21 molecules in colonic tissues from the different groups is shown as bar graphs (n = 10 mice/group; data were analysed by ANOVA followed by Tukey’s HSD post hoc test and are shown as the mean ± SD; a = *p* < 0.05 compared with the NC group; b = *p* < 0.05 compared with the PC group; c = *p* < 0.05 compared with 5-FU monotherapy; d = *p* < 0.05 compared VD_3_ monotherapy; e = *p* < 0.05 compared with Met monotherapy; f = *p* < 0.05 compared with VF dual therapy; g = *p* < 0.05 compared with MF dual therapy; and h = *p* < 0.05 compared with VM dual therapy). (**C**) Localisation of CCND1 and p21 proteins by immunohistochemistry (IHC) in colonic tissues from the different groups (n = 10 mice/group; 20× objective; scale bar = 15 μm).

**Figure 3 cancers-14-01538-f003:**
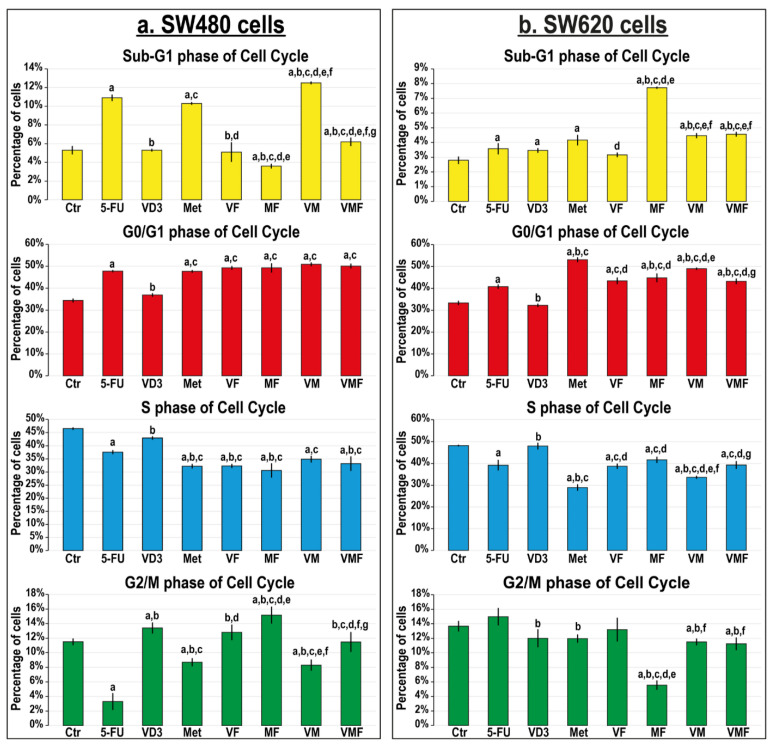
Percentages of cells (mean ± SD) in the different phases of the cell cycle in untreated control cells, and following treatments with 5-fluorouracil, calcitriol, and/or metformin single (5-FU, VD_3_, and Met), dual (VF, MF, and VM), and triple (VMF) therapies for 12 h in the (**a**) SW480 and (**b**) SW620 colon cancer cell lines (n = 3/group; data were analysed by ANOVA followed by Games–Howell post hoc tests and are shown as the mean ± SD; a = *p* < 0.05 compared with control untreated cells; b = *p* < 0.05 compared with 5-FU monotherapy; c = *p* < 0.05 compared VD_3_ monotherapy; d = *p* < 0.05 compared with Met monotherapy; e = *p* < 0.05 compared with VF dual therapy; f = *p* < 0.05 compared with MF dual therapy; and g = *p* < 0.05 compared with VM dual therapy).

**Figure 4 cancers-14-01538-f004:**
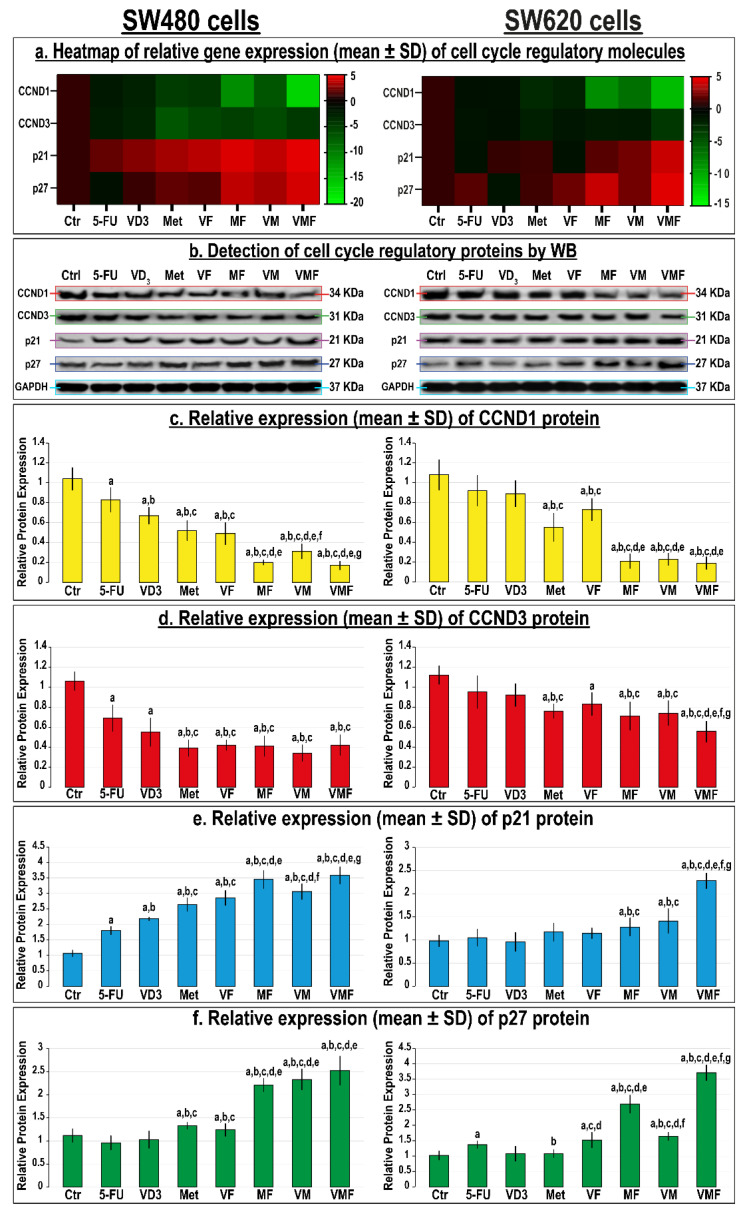
(**a**) Heatmap showing relative mRNA expression (mean ± SD) of CCND1, CCND3, p21, and p27 genes alongside (**b**) detection of their proteins by Western blot and (**c**–**f**) their relative protein expression (mean ± SD), following treatments with 5-fluorouracil, calcitriol, and/or metformin single (5-FU, VD_3_, and Met), dual (VF, MF, and VM), and triple (VMF) therapies for 12 h in the SW480 and SW620 colon cancer cell lines (n = 3/group; data were analysed by ANOVA followed by Games–Howell post hoc tests and are shown as the mean ± SD; a = *p* < 0.05 compared with control untreated cells; b = *p* < 0.05 compared with 5-FU monotherapy; c = *p* < 0.05 compared VD_3_ monotherapy; d = *p* < 0.05 compared with Met monotherapy; e = *p* < 0.05 compared with VF dual therapy; f = *p* < 0.05 compared with MF dual therapy; and g = *p* < 0.05 compared with VM dual therapy).

**Figure 5 cancers-14-01538-f005:**
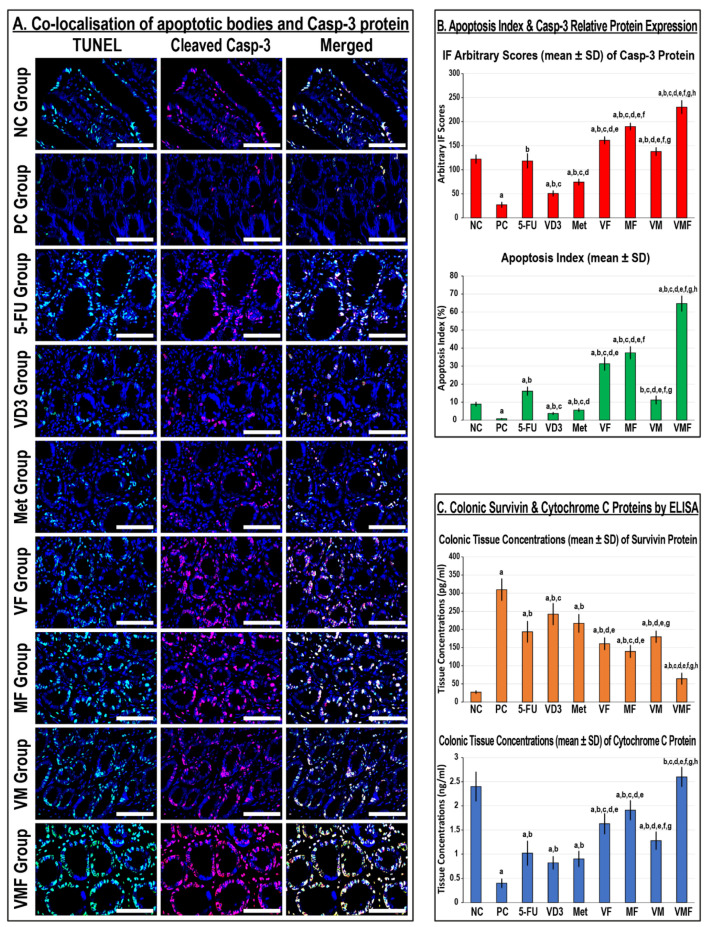
(**A**) Co-detection of apoptotic bodies by TUNEL (green) with cleaved Casp-3 (red) by immunofluorescence in the colonic tissues from all the study groups (n = 10 mice/group; 40× objective; scale bar = 8 µm). (**B**) The relative expression of the Casp-3 protein alongside the apoptosis index and (**C**) the colonic tissue concentrations of survivin and cytochrome C proteins from all groups are displayed as bar graphs (n = 10 mice/group; data were analysed by ANOVA followed by Tukey’s HSD test and are shown as the mean ± SD; a = *p* < 0.05 compared with the NC group; b = *p* < 0.05 compared with the PC group; c = *p* < 0.05 compared with 5-FU monotherapy; d = *p* < 0.05 compared VD_3_ monotherapy; e = *p* < 0.05 compared with Met monotherapy; f = *p* < 0.05 compared with VF dual therapy; g = *p* < 0.05 compared with MF dual therapy; and h = *p* < 0.05 compared with VM dual therapy).

**Figure 6 cancers-14-01538-f006:**
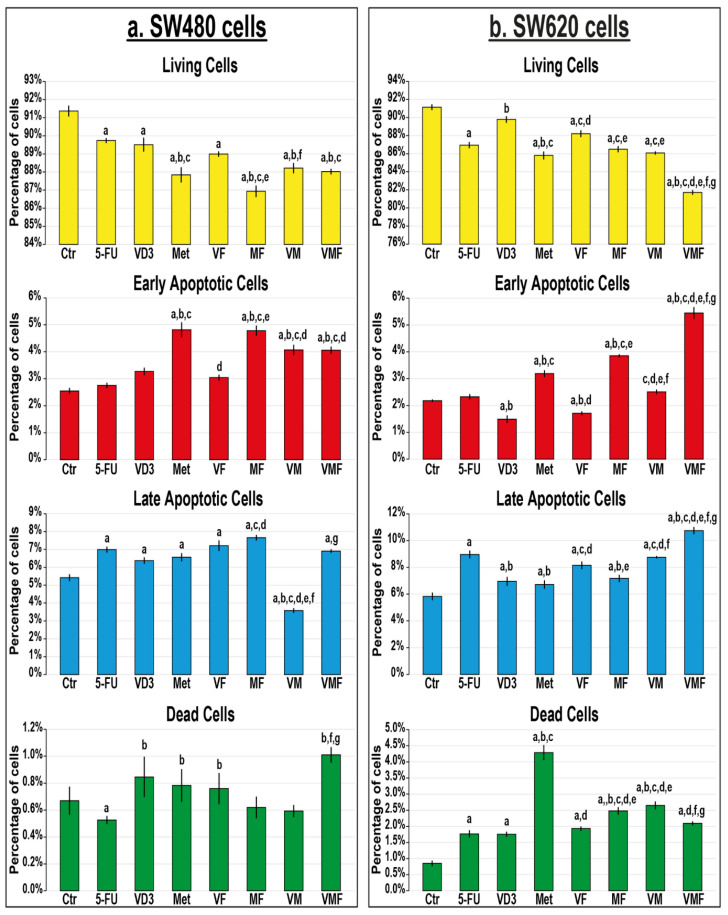
Percentages (mean ± SD) of living, early and late apoptotic, and dead cells in untreated control cells, and following treatments with 5-fluorouracil, calcitriol, and/or metformin single (5-FU, VD_3_, and Met), dual (VF, MF, and VM), and triple (VMF) therapies for 12 h in the (**a**) SW480 and (**b**) SW620 colon cancer cell lines (n = 3/group; data were analysed by ANOVA followed by Games–Howell post hoc tests and are shown as the mean ± SD; a = *p* < 0.05 compared with control untreated cells; b = *p* < 0.05 compared with 5-FU monotherapy; c = *p* < 0.05 compared VD_3_ monotherapy; d = *p* < 0.05 compared with Met monotherapy; e = *p* < 0.05 compared with VF dual therapy; f = *p* < 0.05 compared with MF dual therapy; and g = *p* < 0.05 compared with VM dual therapy).

**Figure 7 cancers-14-01538-f007:**
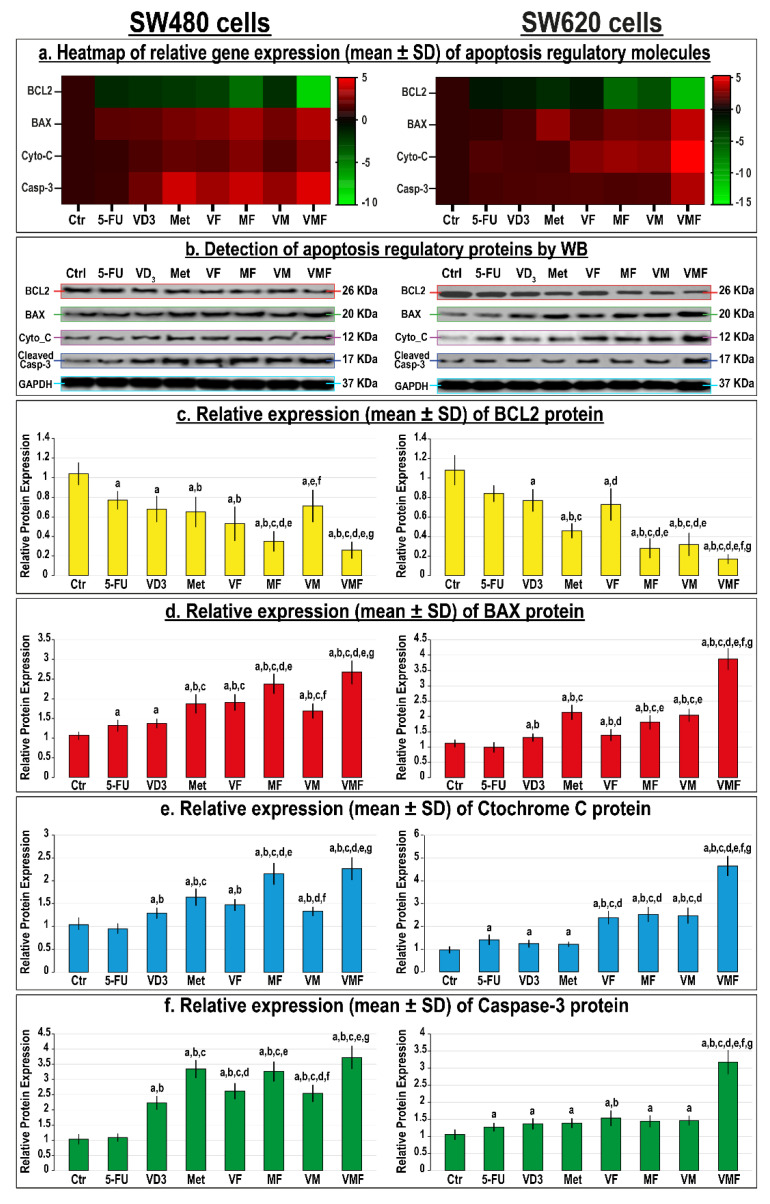
(**a**) Heatmap showing relative mRNA expression (mean ± SD) of BCL2, BAX, cytochrome C, and caspase-3 genes alongside (**b**) detection of their proteins by Western blot and (**c**–**f**) their relative protein expression (mean ± SD) following treatments with 5-fluorouracil, calcitriol, and/or metformin single (5-FU, VD_3_, and Met), dual (VF, MF, and VM), and triple (VMF) therapies for 12 h in the SW480 and SW620 colon cancer cell lines (n = 3/group; data were analysed by ANOVA followed by Games–Howell post hoc tests and are shown as the mean ± SD; a = *p* < 0.05 compared with control untreated cells; b = *p* < 0.05 compared with 5-FU monotherapy; c = *p* < 0.05 compared VD_3_ monotherapy; d = *p* < 0.05 compared with Met monotherapy; e = *p* < 0.05 compared with VF dual therapy; f = *p* < 0.05 compared with MF dual therapy; and g = *p* < 0.05 compared with VM dual therapy).

**Figure 8 cancers-14-01538-f008:**
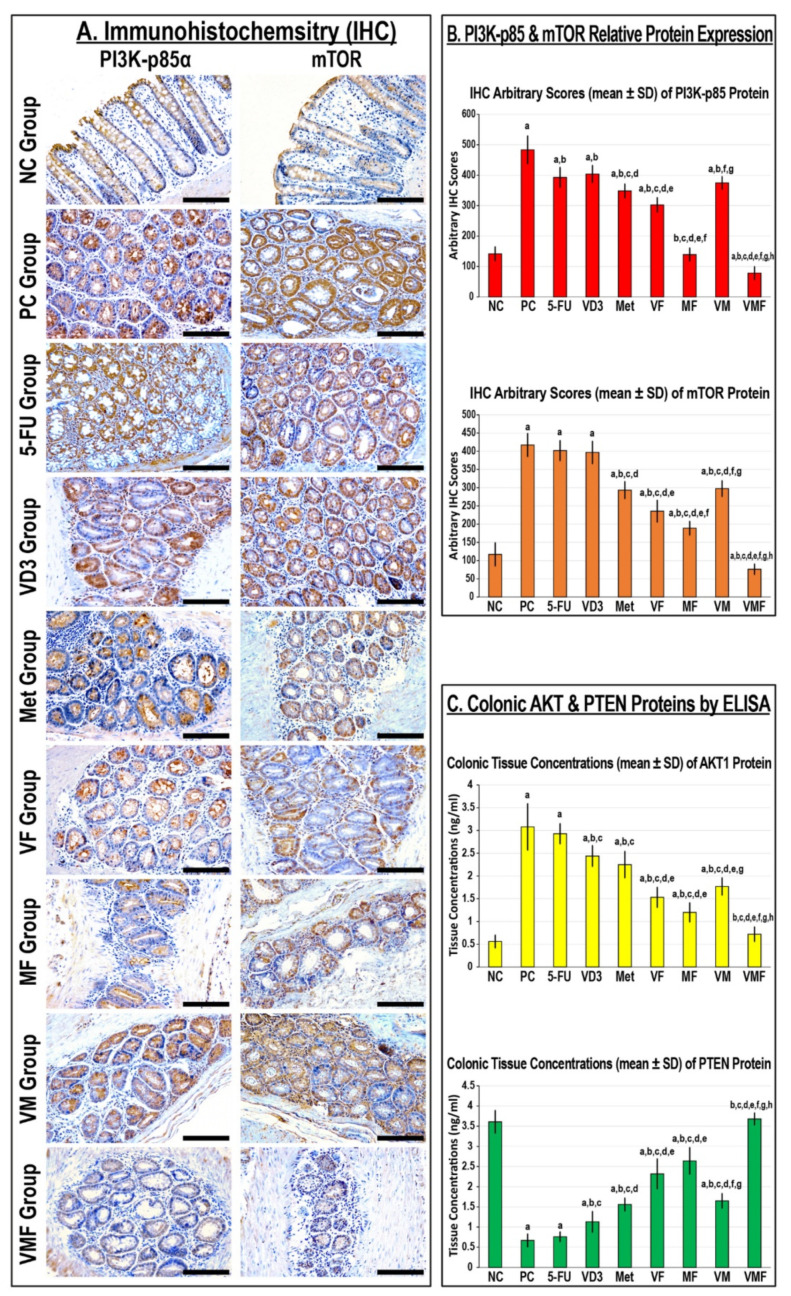
Localisation of PI3K-p85α and mTOR proteins by immunohistochemistry (IHC) in colonic tissues from the different groups (20× objective; scale bar = 15 μm). (**B**) The relative protein expression of PI3K-p85α and mTOR and (**C**) the colonic tissue concentrations of Akt1 and PTEN proteins in the different study groups are displayed as bar graphs (n = 10 mice/group; data were analysed by ANOVA followed by Tukey’s HSD post hoc test and are shown as the mean ± SD; a = *p* < 0.05 compared with the NC group; b *= p* < 0.05 compared with the PC group; c = *p* < 0.05 compared with 5-FU monotherapy; d = *p* < 0.05 compared VD_3_ monotherapy; e = *p* < 0.05 compared with Met monotherapy; f = *p* < 0.05 compared with VF dual therapy; g = *p* < 0.05 compared with MF dual therapy; and h = *p* < 0.05 compared with VM dual therapy).

**Figure 9 cancers-14-01538-f009:**
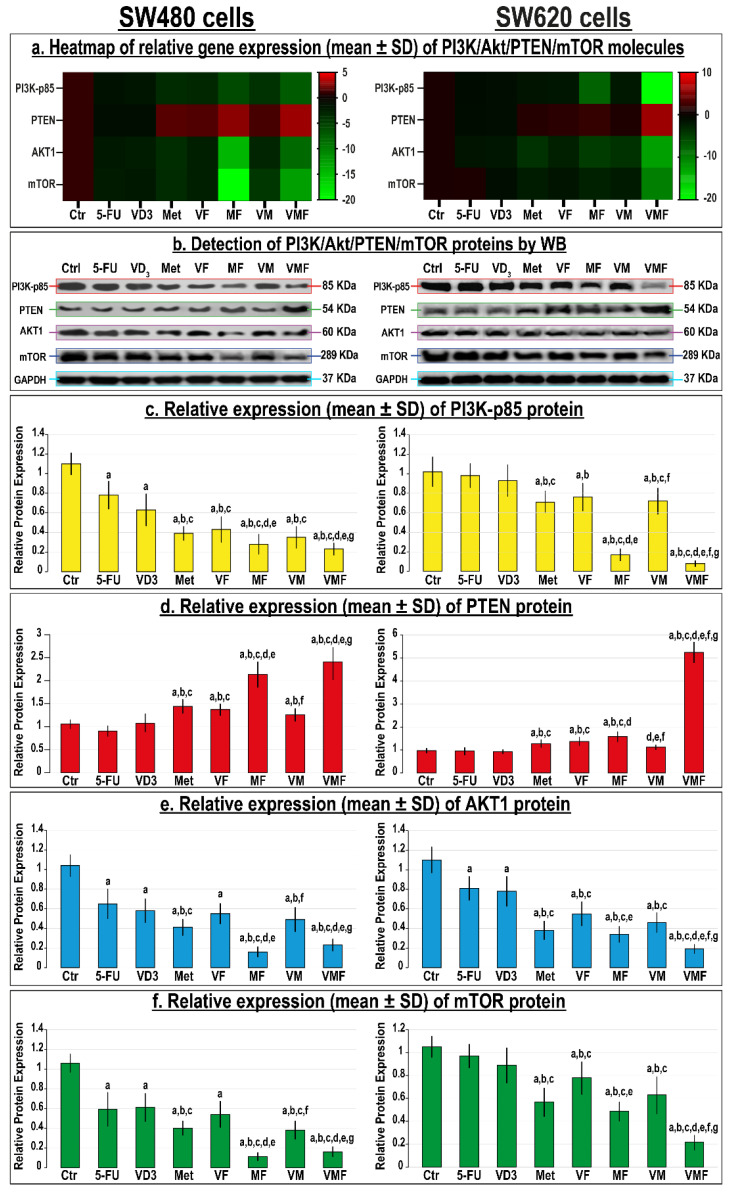
(**a**) Heatmap showing relative mRNA expression (mean ± SD) of PI3K-p85α, PTEN, Akt1, and mTOR genes alongside (**b**) detection of their proteins by Western blot and (**c**–**f**) their relative protein expression (mean ± SD) following treatments with 5-fluorouracil, calcitriol, and/or metformin single (5-FU, VD_3_, and Met), dual (VF, MF, and VM), and triple (VMF) therapies for 12 h in the SW480 and SW620 colon cancer cell lines (n = 3/group; data were analysed by ANOVA followed by Games–Howell post hoc tests and are shown as the mean ± SD; a = *p* < 0.05 compared with control untreated cells; b = *p* < 0.05 compared with 5-FU monotherapy; c = *p* < 0.05 compared VD_3_ monotherapy; d = *p* < 0.05 compared with Met monotherapy; e = *p* < 0.05 compared with VF dual therapy; f = *p* < 0.05 compared with MF dual therapy; and g = *p* < 0.05 compared with VM dual therapy).

## Data Availability

All data generated or analysed during this study are included in this published article (and its Appendix A).

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
