# Peer review of "In Vivo and In Vitro Enhanced Tumoricidal Effects of Metformin, Active Vitamin D3, and 5-Fluorouracil Triple Therapy against Colon Cancer by Modulating the PI3K/Akt/PTEN/mTOR Network"

_cancers, 2022, doi:10.3390/cancers14061538_

Round 1
Reviewer 1 Report
In the manuscript titled, “In vivo and in vitro enhanced tumoricidal effects of metformin, active vitamin D3 and 5-Fluorouracil triple therapy against colon cancer by modulating the PI3K/Akt/PTEN/mTOR network” the authors reported that metformin, active vitamin D3 and 5-Fluorouracil combination therapy has enhanced anti-tumor effect in colon cancer through modulation of PI3K/Akt pathways. The manuscript is well written and would be interesting to cancer researchers. The authors are recommended to make the following changes below.
- In line 129, “Azoxymethane (AOM) was injected for two consecutive weeks….”, specific route of injected was not mentioned.
- In figures 4, 7 and 9, GAPDH western images of SW480 and SW620 cell lines if done at the same set of experiment, can be placed in the bottom of each subfigure instead of repeatedly placing after image of each protein. This will eliminate redundancy.
- Authors are recommended to check whether images of GAPDH in figures 4, 7 and 9 are similar to the corresponding raw images uploaded in the supplemental data.
- Activation of Akt through phosphorylation at Ser473 and Thr308 is critical for cell survival and proliferation in PI3K signaling pathways in colon cancer. The authors are recommended to also include IHC and/or Western data of Akt phosphorylation status in figures 8 and 9.
- Other Akt isoforms, Akt2 and Akt3 are also reported to be upregulated in colon cancer. The author should include data of Akt2/Akt3 in the result section or explain in the discussion section the reason for probing only Akt1.
Author Response
Dear Reviewer,
Thank you for your valuable feedback that will certainly strengthen the scientific outcomes of our manuscript titled: " In vivo and in vitro enhanced tumoricidal effects of metformin, active vitamin D3 and 5-Fluorouracil triple therapy against colon cancer by modulating the PI3K/Akt/PTEN/mTOR network." Please find below detailed answers for your comments.
- In line 129, “Azoxymethane (AOM) was injected for two consecutive weeks….”, specific route of injected was not mentioned.
We have made it clear in the revised version that AOM was injected intraperitoneally for two successive weeks (Page 5; Line 129).
- In figures 4, 7 and 9, GAPDH western images of SW480 and SW620 cell lines if done at the same set of experiment, can be placed in the bottom of each subfigure instead of repeatedly placing after image of each protein. This will eliminate redundancy.
The figures have been amended as requested and a single GAPDH band has been placed at the bottom.
- Authors are recommended to check whether images of GAPDH in figures 4, 7 and 9 are similar to the corresponding raw images uploaded in the supplemental data.
The supplementary figures have been revised and the full images of the used GAPDH matches their corresponding bands for each figure.
- Activation of Akt through phosphorylation at Ser473 and Thr308 is critical for cell survival and proliferation in PI3K signaling pathways in colon cancer. The authors are recommended to also include IHC and/or Western data of Akt phosphorylation status in figures 8 and 9.
The authors appreciate the scientific value of the reviewer’s suggestion. However, this was a phase 1 study that aimed to identify the anti-cancer activities of the proposed therapeutic protocols in relation to the PI3K/Akt/mTOR pathway. Hence, this study measured the total proteins of the molecules of interest, and we plan to fully investigate the actions of each member of the targeted pathway by measuring their phosphorylated isoforms with/without applying specific inhibitors to elucidate their contributions to the enhanced anti-cancer effects for the treatment protocols. Moreover, we have included a statement in the revised discussion under the study limitations that the phosphorylated isoforms of PI3K, Akt and mTOR proteins should be included in future studies (Page 27; Lines 601-604).
- Other Akt isoforms, Akt2 and Akt3 are also reported to be upregulated in colon cancer. The author should include data of Akt2/Akt3 in the result section or explain in the discussion section the reason for probing only Akt1.
The requested clarifications have been added to the revised discussion, as requested (Pages 25; Lines 604-606). Briefly, Akt1 primarily regulates cell survival, and its inhibition is associated with apoptosis. On the other hand, Akt2 is involved in cellular glucose homeostasis, thus we plan to measure it alongside other metabolic markers in future studies.
Reviewer 2 Report
Reviewer comments:
Comments to the Author
This study measured the chemotherapeutic effects of 5-FU, active vitamin D3 (VD3), and/or metformin single/dual/triple regimens as complementary/alternative therapies. Authors observed that all therapies reduced tumour numbers, with 5-FU/VD3/Met the 57 most efficacious regimen. All protocols decreased cell proliferation markers, inhibited PI3K/Akt/mTOR molecules, increased pro-apoptotic molecules with apoptosis index, and 5-FU/VD3/Met revealed the strongest effects.
Due to using three drugs in combination, this study brings novelty to the field to treat colon cancer by applying combinatorial strategies of 5-FU, active vitamin D3 (VD3), and/or metformin against tumor and helps enhanced survival benefits.
The design of the manuscript is interesting while authors need to explain the selection of doses for the combinatorial studies. The experimental designing is in line, and the data provided was comprehensive. The discussion is also well goes with the results and postulated according to the evidence provided.
Major criticisms
- Whether the dose kinetics was performed for all the drugs used in the study. Please provide the data how doses were selected and especially for the combinatorial strategies. Did authors checked the toxicity parameter after using all these combinations. Please provide explanation for these dose selections.
- The authors chose male mice at 12 weeks without mentioning the period of acclimatisation. Were they acclamatised or not? Given this age, what is the clinical relevance of this choice of age and gender?
Minor criticisms
- Please undergo a thorough check of the manuscript for typographical and grammatical errors.
Author Response
Dear Reviewer,
Thank you for your valuable feedback that will certainly strengthen the scientific outcomes of our manuscript titled: " In vivo and in vitro enhanced tumoricidal effects of metformin, active vitamin D3 and 5-Fluorouracil triple therapy against colon cancer by modulating the PI3K/Akt/PTEN/mTOR network." Please find below detailed answers for your comments.
Major Comments
- Whether the dose kinetics was performed for all the drugs used in the study. Please provide the data how doses were selected and especially for the combinatorial strategies.
Herein, the IC50 of each drug was determined following 24h treatment in vitro, as stated in the materials and methods (Page 8; Lines 218-226). The MTT data are included as Supplementary figure 2. Subsequently, the single, dual, and triple therapy protocols were applied for 12h drugs were used to accurately measure their synergistic/enhanced anti-cancer actions related to cell cycle progression and apoptosis alongside the gene and protein expression of their regulatory molecules (Page 8; Lines 218-226).
However, we did not measure the kinetics of the used drugs since each one of them is FDA-approved and used in clinical practice. Moreover, the applied doses of VD3 and metformin in animals were equal to the highest daily doses recommended for humans (Page 5; Lines 134-138). Additionally, many diabetic patients use vitamin D3 in combination with oral hypoglycaemic agents, including metformin, with no or minimal side effects reported (Please refer to PMID: 34489860 and PMID: 34239695 for further details). Earlier experimental animal studies have also shown the safety of combining VD3 with metformin against CRC, as mentioned in the introduction of our manuscript (Please check references 32 & 33). Furthermore, each drug as single-therapy, had significant anti-cancer effects, both in vivo and in vitro, which were enhanced by their dual and triple co-therapy approaches, as demonstrated in the results section.
- Did authors checked the toxicity parameter after using all these combinations. Please provide explanation for these dose selections.
We measured liver enzymes, renal function parameters, serum total protein, and bilirubin, and the results following the different treatment protocols were comparable to the negative control group. The results were included with the original submission as Supplementary Table 3. Hence, we believe that the used doses of the drugs of interest, either single or combined, were safe and not associated with adverse reactions.
- The authors chose male mice at 12 weeks without mentioning the period of acclimatisation. Were they acclamatised or not? Given this age, what is the clinical relevance of this choice of age and gender?
We have made it clear in the revised version that the animals were acclimatised for one week prior to CRC induction by AOM (Page 5; Line 129).
The development of colon adenocarcinoma occurs 15 weeks following induction with AOM. Moreover, the risk of developing CRC in human correlate positively with age, and we used mice of 12 weeks of age to represent 30-40 years old humans (PMID: 31891723). In relation to gender, in vivo experimental research commonly utilises male animals to avoid the potential effects of variations in ovarian sex steroids during oestrus cycle. Ovarian sex steroids have also been shown to protect against CRC development and using female mice would mandate performing ovariectomy to avoid the anti-cancer effects of estrogen and progesterone (PMID: 32451467; PMID: 35201467; PMID: 30064198).
Minor comments
- Please undergo a thorough check of the manuscript for typographical and grammatical errors.?
The manuscript has been thoroughly revised by Dr Aslam, who is a native English speaker, and all corrections have been highlighted in the revised version.
Reviewer 3 Report
In this manuscript, the authors have investigated the anti-tumor effect of metformin, active vitamin D3 and 5-FU combined treatment in colon cancer both in vitro and in vivo. Briefly, the authors examined the anti-tumor effects of metformin (M), active vitamin D3 (V) and 5-FU (F) and their combination in two colon cell lines and in mouse model. They also investigated their possible mechanisms by examining several markers. Overall, the authors suggested the tri-combined treatment of VMF for colon cancer treatement. However, there are some remaining questions to be answered:
1, In SW480 and SW620 cells, although VMF group showed similar effect, VM, MF, VF groups had various response in two cell lines instead. For example, VM group has highest Sub-G1 phase cells in SW480, while MF group has highest in SW620. Could the authors discuss why different cell lines had various response to treatment groups?
2, Fig 2A, missing y-axis title for first panel.
3, Fig 9C, missing grid for first right panel.
4, Since active VD3 was used in the combined treatment, could the authors discuss about the diet supplement for treatment? Is daily supplement VD3 enough for colon cancer patients?
Author Response
Dear Reviewer,
Thank you for your valuable feedback that will certainly strengthen the scientific outcomes of our manuscript titled: " In vivo and in vitro enhanced tumoricidal effects of metformin, active vitamin D3 and 5-Fluorouracil triple therapy against colon cancer by modulating the PI3K/Akt/PTEN/mTOR network." Please find below detailed answers for your comments.
- In SW480 and SW620 cells, although VMF group showed similar effect, VM, MF, VF groups had various response in two cell lines instead. For example, VM group has highest Sub-G1 phase cells in SW480, while MF group has highest in SW620. Could the authors discuss why different cell lines had various response to treatment groups?
The requested explanations related to the in vitro therapeutic efficacies of the VM, and MF dual tprotocols have been added (Page 26; Lines 574-579).
- Fig 2A, missing y-axis title for first panel.
The Y axis title has been added in the revised figure.
- Fig 9C, missing grid for first right panel.
The grid has been added in the revised figure.
- Since active VD3 was used in the combined treatment, could the authors discuss about the diet supplement for treatment? Is daily supplement VD3 enough for colon cancer patients?
A new paragraph that discusses the potential anti-cancer actions of VD3 supplement has been added in the revised manuscript (Page 26; Lines 564-569). Although dietary VD3 has been reported to protect against CRC development, the anti-cancer activities appear to be dose-dependent. Hence, we suggested that future studies should compared the tumoricidal activities of VD3 using low, intermediate, and high doses to reveal the best protective/therapeutic strategies.
Round 2
Reviewer 2 Report
Authors have incorporated the suggestions and this form of manuscript is acceptable.